# A Cooperative Hunting Method for Multi-AUV Swarm in Underwater Weak Information Environment with Obstacles

**Zhenyi Zhao [1], Qiao Hu [1,2,*], Haobo Feng [1], Xinglong Feng [1] and Wenbin Su [1,2]**

1    School of Mechanical Engineering, Xi'an Jiaotong University, Xi'an 710049, China
2    Shaanxi Key Laboratory of Intelligent Robots, Xi'an Jiaotong University, Xi'an 710049, China
*    Correspondence: hqxjtu@xjtu.edu.cn

**Abstract:** Cooperative hunting is a typical task that reflects the intelligence level of a swarm. For the complex underwater weak information environment with obstacles, a problem description of the multi-autonomous underwater vehicle (AUV) cooperative hunting task is given, considering the influencing factors, including underwater obstacles, AUV sensing interaction range, and target escape strategy. A hybrid adaptive preference method based on improved artificial potential fields (HAP-IAPF) is proposed. Then the strategies of obstacle avoidance and hunting are designed separately according to the task requirements. The adaptive weight control unit is used to adjust the preference strategy. The multi-AUV cooperative hunting in dynamic obstacle underwater environments under weakly connected conditions are achieved. In order to prove the effectiveness of the proposed algorithm, simulation results compared with the traditional artificial potential field method and the optimized artificial potential field method are given in this paper. The results show that the proposed method is robust and effective in different environments.

**Keywords:** multi-AUV; cooperative hunting task; obstacle avoidance; dynamic environment





## 1. Introduction

Swarm intelligence behaviors in nature, such as ant swarm [1], bird swarm [2], and fish swarm [3], have attracted the attention of researchers. The multi-agent system composed of intelligent agents represented by intelligent robots [4] is an important research platform for people to study the emergence of intelligent swarm behavior in nature. Multi-agent cooperative hunting is a representative mission that reflects the intelligence of multi-agent systems. Technologies for cooperative hunting missions can provide support for other missions such as search [5], interception, formation change [6], and cooperative transport [7]. It can be widely used in counter-terrorism, security, military, and other fields. The research content involved in the cooperative hunting task includes mathematical modeling of swarms, individual movement modes, information interaction between individuals, intelligent decision-making, task evaluation indicators, etc. The technology involved in robot swarm intelligence is various and complex. Therefore, researchers often evaluate the performance of swarm systems by the completion of the cooperative hunting task [8].

There have been many researchers completing some remarkable studies on some key technologies in the cooperative hunting problem. In geometry-based methods, [9] designed a rigid formation control law using relative position estimates obtained from distance information, and [10] performed formation control based on more readily available azimuth information. But these approaches perform well in some specific static environments rather than dynamic environments. Many studies [11–13] have used grid-based methods to solve the problem of swarm cooperation. Although the solving process is fast and the simulation is easy to approach, the resulting path is discontinuous, and because of the heavy reliance on global information, it is usually not easy to find the heuristic rule in a real-world environment. Sampling-based algorithms are popular in multi-agent motion planning. A

local sampling-based motion planner with a Bayesian learning scheme is studied in [14]. By learning from past observed information, this method can sample regions with a higher probability to form trajectories in narrow areas. However, the method may not produce available paths in a complicated environment because of limited state sampling space. A multi-path planning method for a robot swarm named adversarial RRT* is presented in [15]. The proposed method considers both path cost and a measure of predicted deceptiveness in order to produce a trajectory with a low path cost that still has deceptive properties. However, the resulting trajectory is suboptimal and not curvature continuous. Recently, methods based on artificial intelligence (AI) algorithms have attracted a lot of attention from researchers, ref. [16] proposed a proximal policy optimization (PPO)-based algorithm by taking advantage of reinforcement learning features to reduce computational cost in the reinforcement learning method. Other research [17] studied cooperative control problems of swarming systems in unknown dynamic environments. The swarm agents are required to move in distributed manners with the reference trajectory, which is determined by a virtual dynamic leader. The control policy is modeled by neural networks and updated online. The AI-based methods can usually get a solution quickly, but training a neural network requires a lot of learning data, which is not effective in the case of limited data samples, and may also lead to poor robustness of the obtained strategy. In addition, the artificial potential field (APF)-based methods [18–20] are widely used in distributed swarm tasks because of its advantages in avoiding obstacles and generating paths in real time; however, the solutions may be trapped in a local minimum point sometimes. For the path oscillation caused by the local minimum point problem, ref. [21] proposed an optimized artificial potential field (OAPF) method to ensure the global minimum point is near the target point. Furthermore, the path is smoothed by a step length adjustment unit.

All the strategies discussed above have been studied for swarm robots on the ground or above the water level, which provides stable communication and easy access to information. Little research has been conducted on the problem of swarm cooperative hunting task underwater. Autonomous surface vehicles (ASVs) have attracted a lot of attention as representatives of surface vehicles. The literature [22,23] investigated the ASVs formation path planning problem using the angle guidance fast marching square method developed for operation in dynamic and static environments. A formulation based on closed metric graphs and the application of a multi-objective genetic algorithm was proposed in [24] to provide monitoring solutions for a variable number of ASVs. Based on the constructed bionic swarm pattern and potential function, the swarm velocity guidance with self-organization and collision avoidance was developed in [25] to guide ASVs. As one of the representative platforms of intelligent underwater vehicles, autonomous underwater vehicles (AUVs) are widely used in underwater operations because of their functions of detection, communication, intelligent decision-making, and control. Although AUVs operating in shallow water are close to ASVs operating on the surface in terms of motion models, they are under different environmental communication conditions, and thus the AUV cooperative hunting problem has more constraints. In the multi-AUVs cooperative hunting problem, the multi-AUV system works in an unknown underwater environment, which brings a new challenge for detection, communication, and control of the underwater swarm. Due to the complexity of the underwater environment, the AUV sensing range is very limited and accurate global information cannot be obtained; moreover, obstacle avoidance behavior must be considered in cooperative hunting. Study [26] modeled the AUV hunting problem and used a method to construct a map with complete information to accomplish the hunting, enabling simulation validation of the underwater hunting strategy. However, the strategy requires too much a priori information. Study [27,28] investigated the multi-AUV hunting problem using a bio-inspired neural network (BNN)-based model and proposed a distance-based negotiation method to assign hunting tasks. Simulation studies were conducted in 2-D and 3-D environments, respectively. The proposed algorithm can capture targets relatively quickly. However, it requires global map information for modeling and has limited practical application. Ni [29] presented a dynamic alliance method based on

bidirectional negotiation strategy and a pursuit direction assignment method based on an improved genetic algorithm. Hunter AUVs will negotiate before action, which can effectively improve the navigation efficiency of hunter AUVs. Cao [30] used a method of negotiation to allocate appropriate desired hunting points for each AUV. The hunter AUVs can surround the moving target as expected, however, in studies [24,25], the hunting teams were formed by negotiation, and due to the difficulty of underwater communication, the amount of information interacted by each vehicle was very limited and a large amount of communication was not possible; therefore, this method is difficult to apply in practice. The paper [31] used generative adversarial network (GAN) iterative training to generate a control law suitable for underwater 3D and jamming environments to achieve the successful hunting of noncooperative targets. Liang [32] proposed a behavior-driven coordination control method for multi-AUV hunting problem based on immune mechanism. The hybrid non-central topology is developed with self-organizational and fault-tolerance features. However, studies in [31,32] are based on simple environments without complex situations such as obstacles in the environment, and the control law obtained by training the control relationship in a specific environment as input to the generated model cannot be applied to other broader situations.

The above methods solve the corresponding problems presented in the literature, but they are not applicable to the multi-AUV swarm cooperative hunting problem under a weak information underwater environment due to the large amount of the a priori information required, the high demand for real-time computational resources, or the poor adaptability to different environments.

The main contributions of this work are summarized below:

(1) In order to apply the proposed method in a real underwater environment, the actual constraints of underwater cooperative hunting tasks are considered. An underwater cooperative hunting task model including underwater static and dynamic obstacles, AUV sensing interaction distance limitation, AUV speed variation, target confrontation strategy, and other influencing factors is established.

(2) In order to achieve the stability of the final formation of AUVs, the formation control function of the encirclement process is proposed, which realizes the effective usage of all the AUVs and improves the stability of the final formation. To solve the local oscillation problem during obstacle avoidance, based on the APF-based method, an obstacle avoidance preference motion control function is proposed to realize the smoothing path of the obstacle avoidance and shorten the path length.

(3) To adapt to the requirements of different stages in the cooperative hunting process, an adaptive weight control unit is designed to adjust the collision-free and hunting strategy weights.

The rest of this paper is organized as follows. In Section 2, the problem statement is described. Later, the strategy of multi-AUV cooperative hunting is put forward in Section 3. Section 4 presents the simulation and analysis. Finally, the conclusions are given in Section 5.

## 2. Problem Statement

This paper studies the cooperative hunting task of multi-AUV in an underwater weak information environment with obstacles. The shallow water environment has more obstacles and other disturbances compared to the deep-sea environment. Therefore, this topic is of research value. Since the two-dimensional plane distance variation between the target and the hunter AUV is much larger than the depth variation of each vehicle in most cases in the actual shallow water region, this paper considers the motion in the two-dimensional plane. The cooperative hunting task is described as follows. In the unbounded two-dimensional space, there exists a hunting swarm consisting of $n$ hunter AUVs and a confrontational target with the same intelligence as a hunter AUV, the initial situation of both sides is shown in Figure 1a. Each hunter AUV starts from the initial position, and after several time steps, the hunting task is completed when all hunter AUVs reach around

the intelligent adversarial target and form an encircling formation; the end state of the task is shown in Figure 1b.

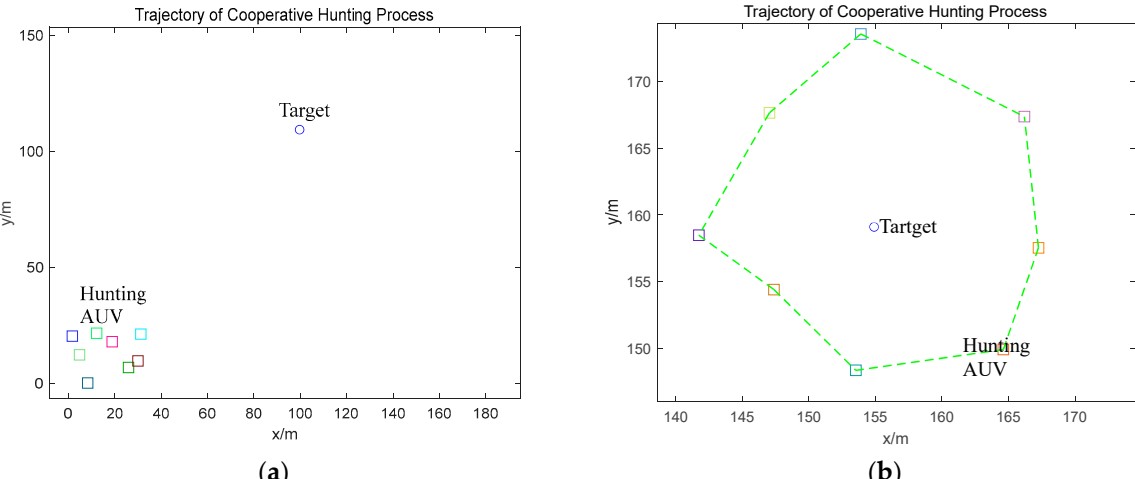

(**a**)         (**b**)

**Figure 1.** (**a**) The initial state of hunter AUV and the target; (**b**) termination state of the cooperative hunting task.

### 2.1. Assumption for Hunter AUV

An omnidirectional mobile AUV platform studied in this paper is shown in Figure 2. The horizontal and vertical propellers provide omnidirectional movement capability for the AUV. The control and sensing cabin include electric field communication and visual detection modules. As a result, the AUV can acquire information within a certain range of its surroundings. In this subsection, some assumptions about the hunter AUVs are listed below, which are necessary to describe the AUV characteristics and the cooperative hunting process.

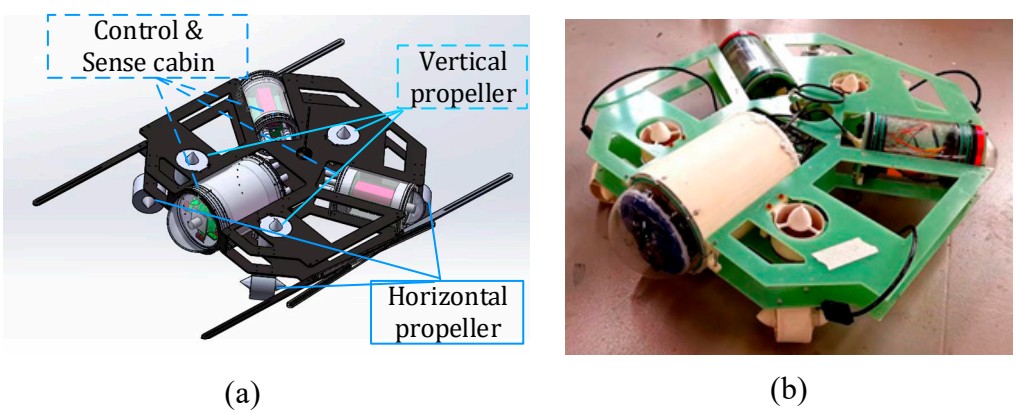

(a)         (b)

**Figure 2.** An omnidirectional mobile AUV: (**a**) structural design; (**b**) real products.

**Assumption 1.** *Kinematics: The AUV has omnidirectional movement capability. For the convenience of description and calculation, the AUV shape is neglected and reduced to a moving mass during the study. Its kinematics is described as:*

$$\begin{cases} \dot{\rho}_i(t) = V_i(t) \\ \dot{V}_i(t) = a_i(t) + f_i(t) \\ |a_i(t)| \leq a_{\max} \\ V_i(t) \in [V_{\min}, V_{\max}] \end{cases} \tag{1}$$

*where, $\dot{\rho}_i(t) \in R$ denotes the two-dimensional coordinate of the ith AUV in the coordinate system at time t, and $\dot{V}_i(t) \in R^2$ denotes the velocity of its corresponding position with a value between the maximum velocity $V_{\max}$ and the minimum velocity $V_{\min}$ of the AUV. $a_i(t)$ denotes the acceleration control variable input by the motion controller, whose absolute value is less than or equal to the maximum acceleration $a_{\max}$. $f_i(t) \in R^2$ denotes the external disturbance, which is a bounded function with an unknown boundary.*

**Assumption 2.** *Priority information: Since the target search problem belongs to another study, the target search is considered to be completed in the roundup problem discussed in this paper, which means that the hunter AUV has a priori information about the initial position $(X_{T0}, Y_{T0})$ of the target. The target location information $(X_T, Y_T)$ can be broadcast through the communication system within the communication distance. The number of all AUVs involved in the mission is known.*

**Assumption 3.** *Sensory information: A single AUV can only obtain position information within a circular area of radius L around it through the electric field-WIFI detection and communication system as follows.*

Friendly neighboring AUV location information:

$$\Omega_i(t) = \left\{ j \in [1,n] \,\middle|\, \|\rho_i - \rho_j\| \le L \right\} \tag{2}$$

where, $n$ denotes the number of all hunter AUVs, and $\rho_j$ denotes the position coordinate of the friendly AUV at the current moment.

Obstacle information available is:

$$\begin{cases} \rho_o(t) = (X_o, Y_o) \\ \dot{\rho}_o(t) = V_o(t) \end{cases} \tag{3}$$

where $\rho_o(t)$ and $V_o(t)$ denote the position and velocity of the obstacle $O$ at moment $t$, respectively.

Target information available is:

$$\begin{cases} \rho_T(t) = (X_T, Y_T) \\ \dot{\rho}_T(t) = V_T(t) \end{cases} \tag{4}$$

where $\rho_T(t)$ and $V_T(t)$ denote the position and velocity of the target at moment $t$, respectively.

### 2.2. Strategy for Intelligent Target

In this subsection, some assumptions about the adversarial targets are listed below, which are used to characterize the intelligent targets.

**Assumption 1.** *Only one intelligent target is hunted by multi-AUV in the paper. The target has similar decision-making and perception capabilities as the hunter AUV, and the target perceives a circular area of radius $r_T$ around it.*

**Assumption 2.** *When the target finds himself being attacked by AUVs, it will immediately start to escape. Its speed and movement direction will be affected due to the hunting influence of AUV. The target's motion strategy is: when there is no enemy target within the sensing range, it moves in a random direction with the cruising speed $V_{T1} < V_{T2}$. When there is a threat, such as a hunter AUV within the sensing range of the target, it escapes in the direction away from the threaten with the fleeing speed $V_{T2}$. The maximum velocity $V_{T2}$ of the moving target is less than the maximum velocity of the AUV.*

## 3. Methods

The APF-based method is a common method for AUV local path planning problem, which requires low communication and detection information for hunter AUV, with less computational complexity and high real-time performance. However, the traditional APF-based method lacks the distinction between cooperative friendly neighbors when applied to multi-AUV cooperative hunting task, and it is easy to fall into local optimum leading to path oscillation or even not reachable to expected area. In this section, we propose a hybrid adaptive preference method based on the improved artificial potential fields (HAP-IAPF) method to solve the problem of multi-AUV cooperative hunting in an underwater weak information environment with obstacles. The flow of the multi-AUV hunting process based on HAP-IAPF is shown in Figure 3.

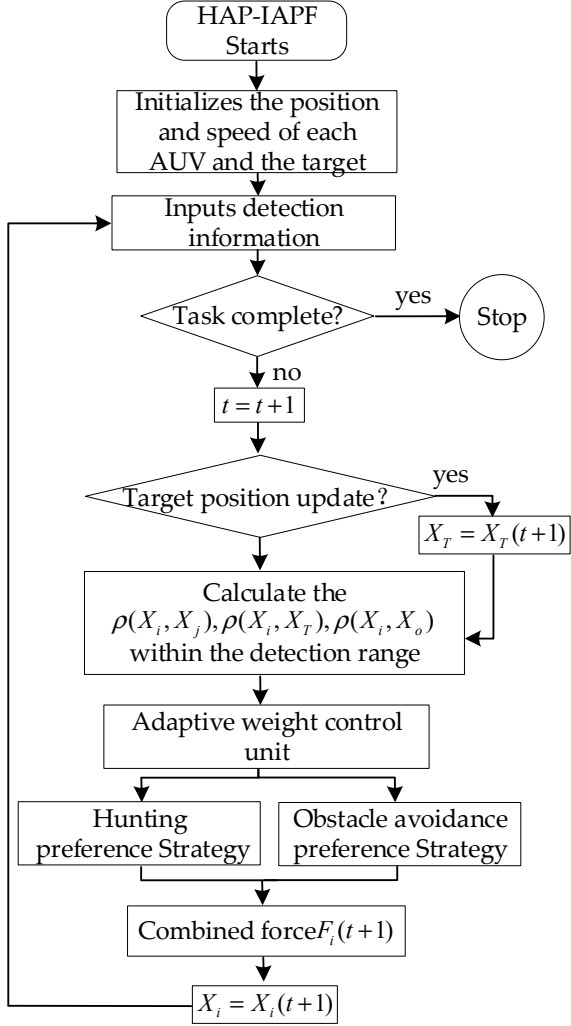

**Figure 3.** Flow chart of HAP-IAPF method.

### 3.1. APF-Based Method

In the traditional APF-based method, the agent is influenced by the potential field of the current position to decide the next movement plan, usually from a high potential energy region to a low potential energy region. The potential field $U(X)$ of position $X$ in the environment is defined as follows:

$$U(X) = U_a(X) + U_r(X) \tag{5}$$

where $U_a(X)$ denotes the attractive potential field generated by the target at position $X_T$, and $U_r(X)$ denotes the repulsive potential field generated by the obstacle. The combined potential field at this point is generated by the superposition of the attractive and repulsive fields, and the negative gradient vector of the combined potential field is taken to control the motion of the AUV.

The attractive potential field is expressed as:

$$U_a(X) = \frac{1}{2}k_1\rho^2(X, X_T) \tag{6}$$

where $k_1$ denotes the gravitational gain parameter and $\rho(X, X_T)$ denotes the Euclidean distance from the current coordinate $X$ to the target position $X_T$.

The attractive force at the corresponding coordinates is:

$$F_a(X) = -\nabla U_a(X) = k_1\rho(X, X_T) \tag{7}$$

The repulsive potential field is expressed as:

$$U_r(X)\begin{cases} \frac{1}{2}k_2\left(\frac{1}{\rho(X,X_o)} - \frac{1}{\rho_0}\right)^2 & 0 \le \rho(X, X_o) \le \rho_0 \\ 0 & \rho(X, X_o) > \rho_0 \end{cases} \tag{8}$$

where $k_2$ denotes the gravitational gain parameter, $\rho(X, X_o)$ denotes the Euclidean distance from the current coordinate $X$ to the obstacle position $X_o$, and $\rho_o$ is a positive number that represents the safe distance between the AUV and the obstacle.

The repulsive force at the corresponding coordinates is:

$$F_r(X) = -\nabla U_r(X)\begin{cases} k_2\left(\frac{1}{\rho(X,X_o)} - \frac{1}{\rho_0}\right) \\ \cdot \frac{1}{\rho^2(X,X_o)}\frac{\partial \rho^2}{\partial X} & 0 \le \rho(X, X_o) \le \rho_0 \\ 0 & \rho(X, X_o) > \rho_0 \end{cases} \tag{9}$$

*3.2. Strategy of Hunting Preference*

The attraction of the target to the hunter AUVs in the traditional APF-based method decreases linearly and tends to zero with decreasing distance, which may lead to the fact that the hunter AUVs are only influenced by the repulsive force between agents at last, which is unfavorable to the stability of final formation. In order to solve this problem and to make the swarm of hunter AUVs quickly and stably surround the target in the final stage, a hunting preference strategy is proposed.

The target attractive function is:

$$F_T(X_i) = [R_T - \|\rho(X_i, X_T)\|] \cdot \frac{1}{\|\rho(X_i, X_T)\|} \cdot \rho(X_i, X_T) \tag{10}$$

where, $R_T$ is the hunting convergence parameter and $\rho(X_i, X_T)$ is the Euclidean distance between the target and the $i$th AUV.

The force function of $i$th AUV generated by all friendly neighbor AUVs is:

$$F_{ij}^1(X_i) = \begin{cases} \sum\limits_{\substack{j=1 \\ j \ne i}}^{n} \left[\left(\frac{1}{\|\rho(X_i,X_j)\|} - \frac{2\sin(\frac{\pi}{n})}{L}\right) \right. \\ \left. \cdot \frac{1}{\|\rho(X_i,X_j)\|^2} \cdot \rho(X_i, X_j)\right] & \rho(X_i, X_j) \le L \\ 0 & \rho(X_i, X_j) > L \end{cases} \tag{11}$$

where $n$ is the number of all hunter AUVs, and $\rho(X_i, X_T)$ is the Euclidean distance between $i$th AUV and a friendly neighbor $j$th AUV within the sensing range.

*3.3. Strategy of Obstacle Avoidance Preference*

In the traditional APF method, when an individual is affected by both attractive and repulsive forces around an obstacle, there may be a positive or negative change in the combined force at the next moment as the AUV moves. Therefore, the path planned by the AUV may have repeated oscillations, resulting in wasted energy and time. In order to solve this problem and make the multi-AUV swarm pass the obstacle area quickly and stably, an obstacle avoidance preference strategy with improved virtual force is proposed.

The target attractive force is the same as Equation (10), and the force function of $i$th AUV generated by all friendly neighbor AUVs is:

$$
F_{ij}^2(X_i) = \begin{cases} \sum_{\substack{j=1 \\ j \neq i}}^{n} \left[ \left( \frac{1}{\|\rho(X_i, X_j)\|} - \frac{2\sin(\frac{\pi}{n})}{L} \right) \\ \cdot \frac{1}{\|\rho(X_i, X_j)\|^3} \cdot \rho(X_i, X_j) \right] & \rho(X_i, X_j) \leq L \\ 0 & \rho(X_i, X_j) > L \end{cases} \tag{12}
$$

Suppose there are $m$ obstacles or equivalent obstacles within the detection range of the $i$th AUV, the obstacle repulsion is:

$$
F_o(X_i) = \begin{cases} \sum_{j=1}^{m} \left[ \left( \frac{1}{\|\rho(X_i, X_{oj})\|} - \frac{1}{R} \right) \\ \cdot \frac{1}{\|\rho(X_i, X_{oj})\|^2} \cdot \rho(X_i, X_{oj}) \right] & \rho(X_i, X_{oj}) \leq R \\ 0 & \rho(X_i, X_{oj}) > R \end{cases} \tag{13}
$$

where $\rho(X_i, X_{oj})$ is the Euclidean distance between $i$th AUV and the $j$th obstacle within the detection range of the $i$th AUV. $R$ denotes the obstacle rejection action range.

*3.4. Adaptive Weight Control Unit*

The adaptive weight control unit is designed to select the current best strategy based on the state of a hunter AUV. $\rho(X_i, X_{omin})$ is the Euclidean distance between the current AUV and the nearest obstacle within the effective distance. The input of the control unit is:

$$
c_{In} \begin{cases} \frac{\rho(X_i, X_{omin})}{R} & \rho(X_i, X_{omin}) exist \\ 1 & otherwise \end{cases} \tag{14}
$$

The output is the matrix $C = [k_a \ k_b \ k_c]$.
Introduce the function:

$$
\mu(x) = \frac{1}{2} \left[ -\tan\left( x \cdot \frac{\pi}{2} - \frac{\pi}{4} \right) + 1 \right] \ x \in [0, 1] \tag{15}
$$

where,

$$
\begin{cases} k_a = 2\mu(c_{In}) \\ k_b = h_0 \\ k_c = k_b \mu(c_{In}) \end{cases} \tag{16}
$$

where, $h_0$ is a positive constant.

Eventually, the combined force on the hunter AUV is

$$
F_i = \begin{cases} C \cdot \left[ F_T F_{ij}^1 F_O \right]^T & c_{In} = 1 \\ C \cdot \left[ F_T F_{ij}^2 F_O \right]^T & otherwise \end{cases} \tag{17}
$$

When the gravitational force and repulsive force on the individual just cancel, it may cause the AUV to fall into the local optimal value point and cannot move further. Therefore, the rule is set when $F_i(t) = 0$, a small random perturbation $\omega$ that is perpendicular to the direction of the target linkage is added.

## 4. Simulation Results

In order to verify the feasibility and effectiveness of the proposed algorithm, simulations and analyses in a static obstacle environment and dynamic obstacle environment are given, respectively. The performance of the proposed algorithm is compared with the APF-based algorithm and the OAPF method. The simulation platform is a computer with Intel(R) Core(TM) i7-8700 CPU 3.20 GHz which was developed by Intel Inc. in Santa Clara, California, USA, 16 GB memory, and the software is MATALAB R2020a which was developed by MathWorks, Inc. in Natick, Massachusetts, USA.

The general parameters of the simulation are listed in Table 1. The maximum speed of the hunter AUV is set to be greater than the target escaping speed $V_{T2}$. The sensing distance $L$ of the hunter AUV is less than the target sensing distance $r_T$. The obstacle action radius $R$ is smaller than the perception distance of the hunter AUV, which ensures that the hunter AUV can respond to existing obstacles. The values of other parameters were adjusted by observing the system behavior during simulations. We verified that these parameters could possibly influence the AUVs during the task execution but were definitely not decisive in the accomplishment of the task.

**Table 1.** Parameters for simulation.

| Symbol | Description | Value | Units |
|:------:|:-----------:|:-----:|:-----:|
| $t$ | Steps of time | 1 | s |
| $n$ | Number of hunter AUVs | 8 | - |
| $T$ | Maximum steps of simulation | 100 | s |
| $L$ | AUV sensing range radius | 10 | m |
| $V_{T1}$ | Target cruising speed | $[-0.3, -0.5]$ | m/s |
| $V_{T2}$ | Target escaping speed | 1.697 | m/s |
| $V_{\max}$ | Maximum speed of hunter AUVs | 3 | m/s |
| $V_{\min}$ | Minimum speed of hunter AUVs | 0.1 | m/s |
| $X_t$ | Target initial coordinates | $[100, 110]$ | m |
| $r_T$ | Target sensing range radius | 20 | m |
| $R$ | Obstacle influence range | 8 | m |
| $R_T$ | Hunting convergence parameters | 10 | m |
| $h_0$ | Friendly neighbor repulsion constant | 1 | - |
| $\omega$ | Perturbation | 0.1 | - |

### 4.1. Static Obstacle Environment Simulation

In order to verify the effectiveness of the proposed method, a simulation study is performed in a static obstacle environment. The two circular static obstacles with center location coordinates of [80, 70], [80, 90] and a radius of 2 m are near the path that the AUV swarm is expected to pass through. The safety distance between AUV and the center of the obstacle is 3 m.

The HAP-IAPF simulation results are shown in Figures 4 and 5. It is considered that the task is completed when the error between the geometric center position of the formation composed of the hunter AUVs and the target position is less than 1 m.

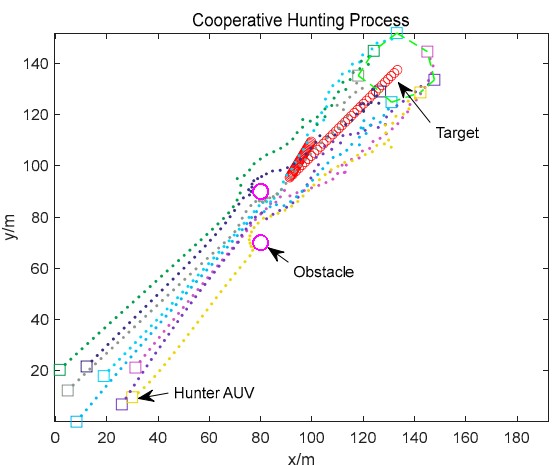

**Figure 4.** Cooperative hunting process in static obstacle environment based on HAP-IAPF method.

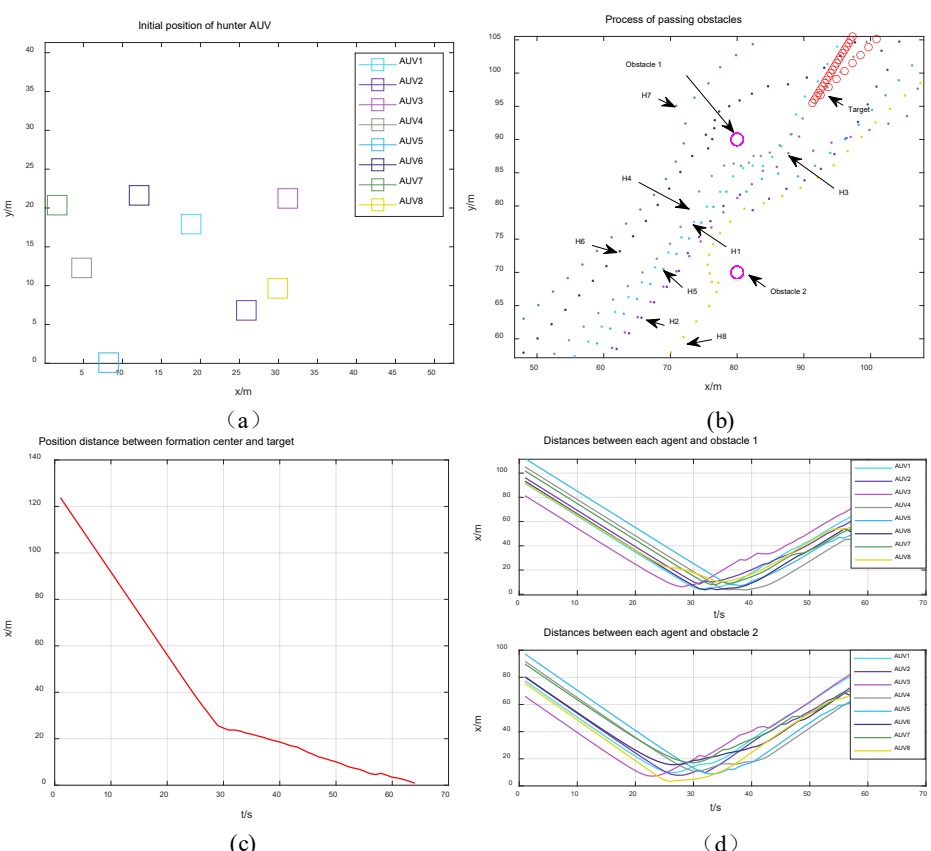

**Figure 5.** HAP-IAPF method in static environment: (**a**) The initial position of hunter AUV; (**b**) process of passing obstacles; (**c**) position distance between formation center and the target; (**d**) distances between each agent and obstacle 1 and 2.

As shown in Figure 4, after starting from the initial position located near the origin (see Figure 5a), the 8 hunter AUVs smoothly pass the area affected by obstacles (see Figure 5b) and finally successfully surround the target. A stable and regular circular formation is formed around the target finally. In Figure 5c, it is shown that the cooperative hunting process is convergent. After 64 steps, the target has been completely surrounded. In Figure 5d, it is shown that the AUV swarm passes through the obstacle-influenced area with a smooth avoidance path, and the closest distance of AUV from the obstacle is 3.24 m, which is successfully kept above the safe distance.

The APF simulation results are shown in Figures 6 and 7.

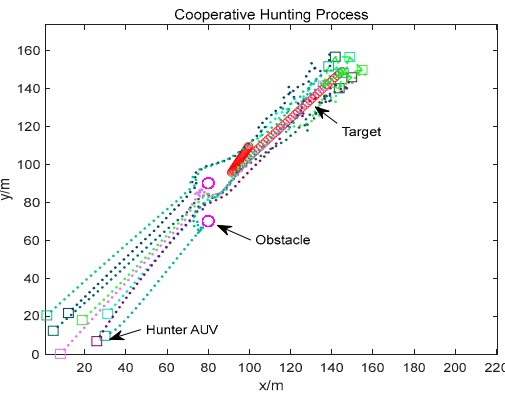

**Figure 6.** Cooperative hunting process in static obstacle environment based on the APF-based method.

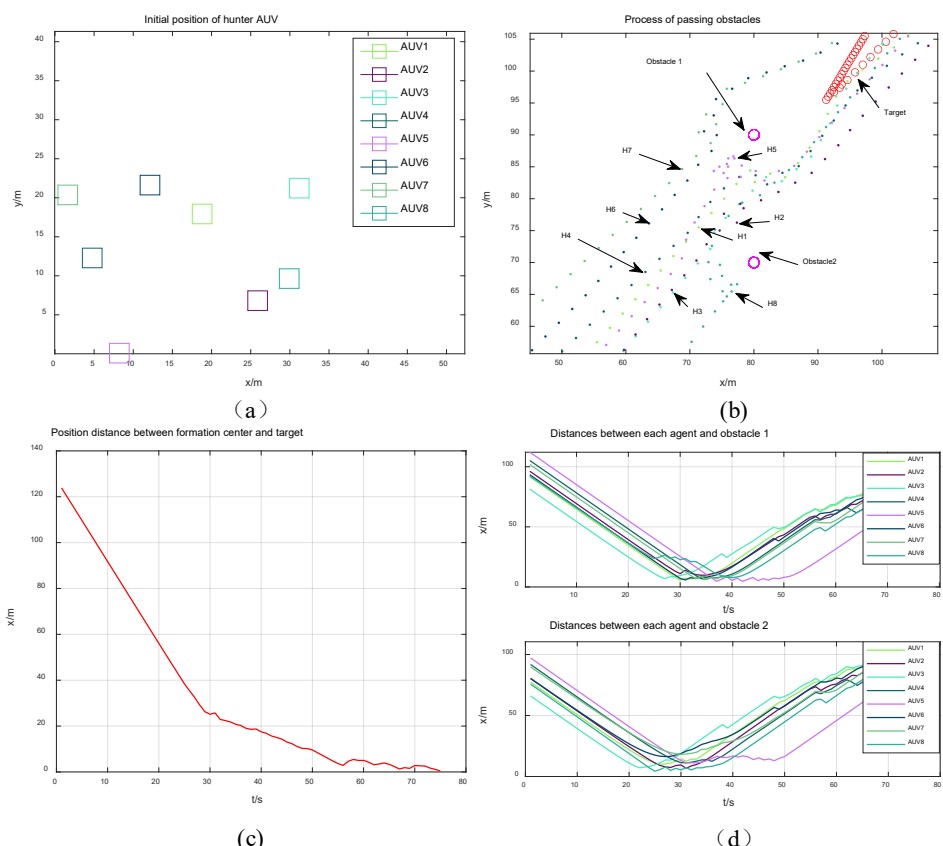

**Figure 7.** APF-based method in static environment: (**a**) The initial position of hunter AUV; (**b**) process of passing obstacles; (**c**) position distance between formation center and the target; (**d**) distances between each agent and obstacle 1 and 2.

As shown in Figure 6, after starting from the initial position located near the origin (see Figure 7a), the 8 hunter AUVs smoothly pass the area affected by obstacles (see Figure 7b) and surround the target finally. The final formation formed around the target is an irregular polygon, and the final formation has a lot of randomness. Figure 7c shows that after 75 steps, the target has been completely surrounded. Figure 7d shows that when the AUV swarm passes through the obstacle-influenced area, there is some oscillation in the avoidance path, which leads to an increase in the avoidance path length. The closest distance of the AUV from the obstacle is 4.30 m, which is successfully kept above the safe distance.

The AUV swarm enters the obstacle influence area at about 20–40 steps. The path length of each AUV during this period is given in Table 2. The average path length of the

HAP-IAPF is 57.4 m, and that of the APF method is 60.4 m. Referring to Figures 5d and 7d, the HAP-IAPF method can plan a smoother and shorter path when avoiding obstacles. Both methods successfully surround the target at 64 steps and 75 steps, respectively, while the proposed method in this paper has a shorter completion time.

**Table 2.** Path length of each AUV during 20–40 steps (unit: m).

| AUV Number | 1 | 2 | 3 | 4 | 5 | 6 | 7 | 8 | Average |
|---|---|---|---|---|---|---|---|---|---|
| HAP-IAPF | 56.4 | 62.8 | 60.8 | 52.0 | 62.5 | 50.8 | 62.9 | 50.7 | 57.4 |
| APF based | 61.1 | 62.2 | 62.0 | 61.6 | 59.7 | 61.1 | 61.2 | 54.5 | 60.4 |

### 4.2. Dynamic Obstacle Environment Simulation

To verify the effectiveness of the proposed method, a simulation study is performed in a dynamic obstacle environment. The initial center position coordinates of two circular dynamic obstacles are [75, 95], [100, 40], respectively, Obstacle 1 and Obstacle 2 make a uniform linear motion with velocities of 0.3 m/s and 1.2 m/s, respectively, and the motion trajectory is shown in Figure 8.

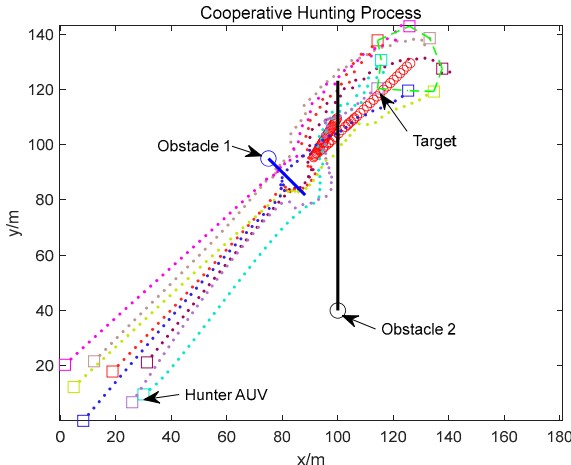

**Figure 8.** Cooperative hunting process based on the HAP-IAPF method in a dynamic obstacle environment.

The HAP-IAPF simulation results are shown in Figures 8 and 9.

As shown in Figure 8, after starting from the initial position located near the origin (see Figure 9a), the 8 hunter AUVs smoothly pass the area affected by obstacles (see Figure 9b) and finally successfully surround the target. A stable and regular circular formation is formed around the target at last. In Figure 9c, it is shown that the target is completed to be surrounded after 63 steps. Figure 9d, shows that the AUV swarm passes through the obstacle-influenced area with a smooth avoidance path, and the closest distance of AUV from the obstacle is 3.11 m, which maintains a safe distance.

The APF simulation results are shown in Figures 10 and 11.

As shown in Figure 10, after starting from the initial position located near the origin (see Figure 11a), the 8 hunter AUVs smoothly pass the area affected by obstacles (see Figure 11b) and surround the target finally. The final formation is an irregular polygon. In Figure 11c, it is shown that the target is completed surrounded after 75 steps. Figure 11d shows that the AUV swarm passes through the obstacle-influenced area with some oscillations in the avoidance path, resulting in an increase in the avoidance path length. The closest distance of the AUV from the obstacle is 4.82 m, which successfully maintains a safe distance or more.

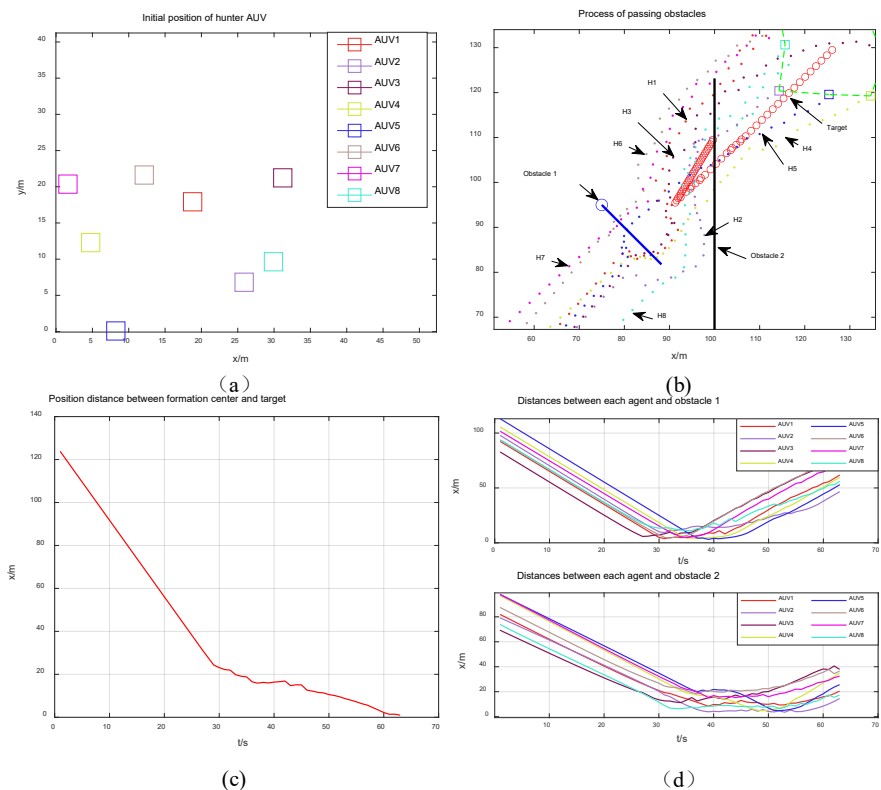

**Figure 9.** HAP-IAPF method in dynamic environment: (**a**) The initial position of hunter AUV; (**b**) process of passing obstacles; (**c**) position distance between formation center and the target; (**d**) distances between each agent and obstacle 1 and 2.

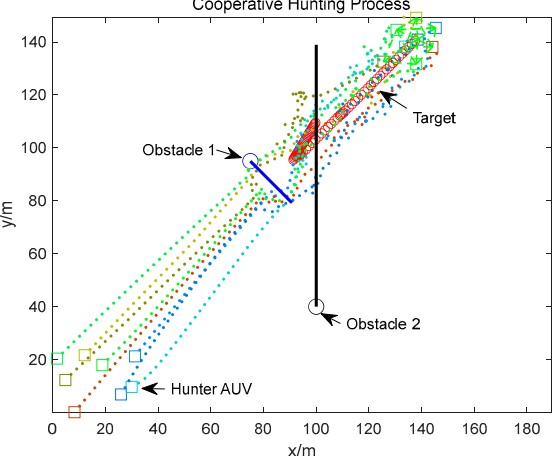

**Figure 10.** Cooperative hunting process based on the APF-based method in a dynamic obstacle environment.

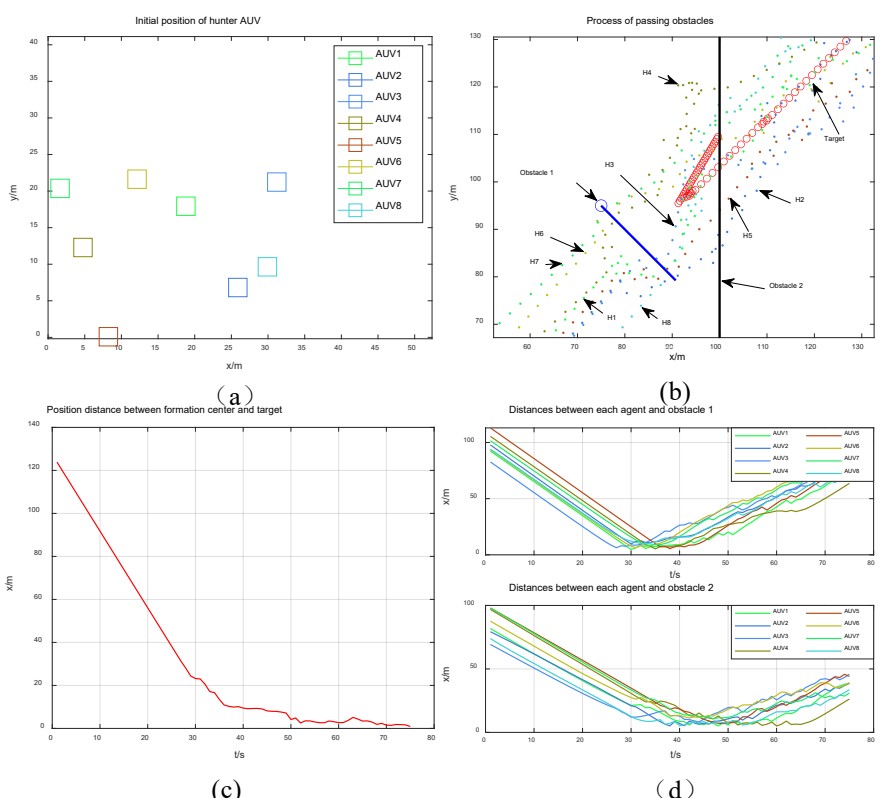

**Figure 11.** APF-based method in dynamic environment: (**a**) The initial position of hunter AUV; (**b**) process of passing obstacles; (**c**) position distance between formation center and the target; (**d**) distances between each agent and obstacle 1 and 2.

The AUV swarm enters the obstacle impact area at about 20–60 steps, and the path length of each AUV is given in Table 3. The average path length is 111.7 m for HAP-IAPF and 118.4 m for APF method, referring to Figures 9d and 11d, it can be concluded that the HAP-IAPF method has a smoother and shorter path during obstacle avoidance. Both methods successfully surround the target at 63 steps and 75 steps, respectively, while the proposed method in this paper has a shorter completion time. Thus, the simulation results indicate the adaptability of the HAP-IAPF method in an unknown dynamic environment.

**Table 3.** Path length of each AUV during 20–60 steps (unit: m).

| AUV Number | 1 | 2 | 3 | 4 | 5 | 6 | 7 | 8 | Average |
|---|---|---|---|---|---|---|---|---|---|
| HAP-IAPF | 111.8 | 98.2 | 119.6 | 107.0 | 107.2 | 117.7 | 118.8 | 113.6 | 111.7 |
| APF-based | 115.4 | 121.0 | 121.5 | 113.6 | 120.0 | 122.2 | 123.0 | 110.4 | 118.4 |

*4.3. Analysis*

The comparison of HAP-IAPF, APF-based, and the OAPF algorithms proposed in [21] is shown in Table 4. After 100 times Monte Carlo simulations, the total calculation time of HAP-IAPF, APF-based, and OAPF are listed. The computation time of the HAP-IAPF method increased by 6.71% and 7.04% compared to APF-based algorithm. It can be seen that both algorithms had a fast response time and similar computation time. The computation time of the OAPF method increased by 44.88% and 50.64% compared to APF-based algorithm. The result shows that the computation time of OAPF algorithm increased significantly. Thus, the OAPF algorithm is not good in real-time, which affects its application in an underwater environment. The average lengths of obstacle avoidance paths in 20–40 s for each algorithm are 57.4 m, 60.4 m, and 59.4 m in the static environment. The average lengths for 20–60 s in the dynamic environment were 111.7 m, 118.4 m, and 108.1 m. HAP-

IAPF decreased by 4.97% and 5.32% in the different environments relative to APF-based, respectively. OAPF decreased by 1.66% and 8.69% in the different environments relative to APF-based, respectively. The path of the OAPF algorithm was the shortest path in the dynamic environment. Because of the local minimum problem, the OAPF algorithm prefers to approach the target point quickly and ignores the effect of moving obstacles. This may lead to an unsafe obstacle avoidance process, therefore, the OAPF is not very adaptable to dynamic environments. Define a change in heading angle of more than 90 degrees as a large deflection. By comparing the changes in headings for each AUV, the average number of large deflections occurred by HAP-IAPF was reduced by 23.42% and 41.94% relative to APF-based in the static and dynamic environments, respectively. It was reduced by 8.44% and 18.18% relative to OAPF in the static and dynamic environments, respectively. The step length adjustment unit in the OAPF algorithm can achieve a certain degree of path smoothing, while the algorithm proposed in this paper is more effective. The completion time of HAP-IAPF was 14.67% and 16% shorter than APF-based in different environments, respectively. It was reduced by 4.48% and 24.10% compared with the OAPF algorithm. This was further evidence that the OAPF algorithm was not quite suitable for cooperative hunting problems in dynamic environments. The computational cost of the method in this paper was low, and the average path length and average time were shorter. Moreover, the formed formation was stable. Therefore, the method proposed in this paper is suitable for cooperative hunting tasks in unknown underwater environments.

**Table 4.** Comparison of algorithms.

| Simulation Environment | Algorithm | Calculation Time (s) | Path Length (m) | Heading Deflections (Times) | Completion Time (s) |
|---|---|---|---|---|---|
| Static Environment | HAP-IAPF | 119.78 | 57.4 | 4.12 | 64 |
| | APF-based | 112.25 | 60.4 | 5.38 | 75 |
| | OAPF | 162.63 | 59.4 | 4.50 | 67 |
| Dynamic environment | HAP-IAPF | 126.00 | 111.7 | 4.50 | 63 |
| | APF-based | 117.71 | 118.4 | 7.75 | 75 |
| | OAPF | 177.33 | 108.1 | 5.50 | 83 |

## 5. Conclusions

In this paper, we propose a HAP-IAPF method to accomplish a cooperative hunting task in an unknown underwater environment. First, we propose a task model that includes underwater static and dynamic obstacles, AUV sensing interaction distance limitation, AUV speed variation, and target confrontation strategy to be as close as possible to the real underwater environment. Then, the final formation stabilization is achieved by designing a hunting preference strategy. By designing the obstacle avoidance preference strategy, the path is smoothed, and the efficiency is improved. Finally, to adapt to the requirements of different stages in the cooperative hunting process, an adaptive weight control unit is designed to adjust the collision-free and hunting strategy weights. We simulated in static obstacle environment and dynamic obstacle environment, respectively. The results show that the proposed method has a low computational cost, shorter average path length, and average time than APF-based and OAPF method. Moreover, the formed formation is stable. Thus, the effectiveness of the proposed method in an unknown underwater environment is proven.

In the future, on the one hand, the influence of distinguishing different speed obstacles or friendly individuals is considered, thus allowing the AUV to adjust its speed and heading autonomously according to different types of obstacles to further improve the adaptability of the strategy. On the other hand, the proposed method is validated experimentally in a real environment using the designed AUV.

**Author Contributions:** Conceptualization, Z.Z. and Q.H.; methodology, Z.Z. and H.F.; software, Z.Z. and H.F.; formal analysis, X.F.; writing—original draft preparation, Z.Z.; writing—review and editing, Z.Z. and Q.H.; visualization, X.F.; supervision, W.S. All authors have read and agreed to the published version of the manuscript.

**Funding:** This work was supported by the Innovation Special Zone Project of China under Grant 193A1111040501 and the Basic Research Project of China under Grant JCKY2020110C074.

**Institutional Review Board Statement:** Not applicable.

**Informed Consent Statement:** Not applicable.

**Data Availability Statement:** Not applicable.

**Conflicts of Interest:** The authors declare no conflict of interest.

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
