# Peer review of "A Cooperative Hunting Method for Multi-AUV Swarm in Underwater Weak Information Environment with Obstacles"

_jmse, doi:10.3390/jmse10091266_

Round 1

Reviewer 1 Report

The authors started with a survey of cooperative guidance on the ground and in the air as the support for their main topic related to underwater. However they did not mention surface autonomous vehicles which would be a closer problem, so this should be added, where examples could be found in:

https://doi.org/10.1016/j.oceaneng.2019.04.098 

https://doi.org/10.1016/j.apor.2016.06.013

The authors want to deal with underwater vehicles and thus they need to consider trajectories in 3d space but they seem to deal basically with a 2D problem. Many of the plots of results show basically the x-y trajectories.

Furthermore, in lines 162- 163 it is mentioned that the vehicles operate in a two-dimensional coordinate system.

In lines 195-196 it is indicated that the target is perceived in a circular area of a given radius instead of perceiving in a spherical volume of a given radius.

Therefore it appears that the authors are dealing with a swarm of surface vehicles, which operate in one plane instead of a swarm of underwater vehicles that can move in three dimensions. In this case, the titles and narrative should be adjusted to deal with that problem.

In Fig 4 it should be Obstacle instead of Obstacal

Reviewer 2 Report

The paper presents a modified approach for multi-swarm AUV hunting in an environment with obstacles. Authors use improved artificial potential fields in order to design obstacle avoidance and hunting strategies. In general the paper is readable, nevertheless some comments are the following:

1.-In the abstract, authors state that "The multi-AUV cooperative hunting in dynamic obstacle underwater environments under weakly connected is achieved with low communication, low computational cost and high real-time performance". The use of adjectives such as "low" might lead to controversies, please clarify. The same with the words "good robustness".

2.- What are the main differences with the multi-swarm hunting problem for wheeled robots and AUV robots?

3.- Authors state some contributions of this work in lines 116-130. However this reviewer disagrees that simulation is an original contribution. Moreover,  authors claim "In order to achieve the stability of the final formation of AUVs, the formation control function of the encirclement process is proposed, which realizes the effective usage of all the AUVs and improves the stability of the final formation. To solve the local oscillation problem during obstacle avoidance based on the APF-based method, an obstacle avoidance preference motion control function is proposed to realize the smoothing path of the obstacle avoidance and shorten the path length." This a strong claim that must be supported by an Lyapunov stability analisys, therefore authors are encouraged to devise it.

4.- Figure 1b must be improved.

5.- All equations typograpy must be Homogenized. For example, equation (1) uses lower case letters but in figure 3 and equations 5-17 upper cases are used. Pleas fix.

6.- In line 303 authors state: "The values of other parameters were experimentally tuned by observing the system behavior during simulations", what does "experimenally" means? Is it that they were obtained from the physical robot? Please, clarify.

7.-All the simulations results sections is hard to read. May be a simulation video might help to better understand the results. Table 2 compares HAP-IAPF vs APF; results might lead to conclusions that HAP-IAPF is a better strategy. Nevertheless, this is true only for the tested simulations and more general results need to be obtained. 

8.- All the above considerations will help authors to get sharpen conclussions.

Round 2

Reviewer 1 Report

The authors have confirmed that their work is in 2D. In this case my earlier message is that a 2D formulation is not specific of AUV, which can move in a 3D space.  On the contrary Autonomous Surface vehicles move in 2D space.  Therefore, it appears that the work that the authors are presenting for a AUVs is equally applicable to USVs, and this should be made clear in the paper and a proper literature review on the work done in USVs is missing

Author Response

Thank you very much for your comments and suggestions on our manuscript entitled “A cooperative hunting method for multi-AUV swarm in underwater weak information environment with obstacles” (ID: jmse-1853109).

We have made the following changes.

  1. We added a summary of the research progress on ASVs in lines 77-86. “Autonomous surface vehicles (ASVs) have attracted a lot of attention as representatives of surface vehicles. The literature [22,23] investigated the ASVs formation path planning problem using the angle guidance fast marching square method developed for operation in dynamic and static environments. A formulation based on closed metric graphs and the application of a multi-objective genetic algorithm is proposed in [24] to provide monitoring solutions for a variable number of ASVs. Based on the constructed bionic swarm pattern and potential function, the swarm velocity guidance with self-organization and collision avoidance is developed in [25] to guide ASVs.”
  2. We added a comparison of ASVs and AUVs operating in shallow water, including a summary of similarities and differences, in lines 88-91. “Although AUVs operating in shallow water are close to ASVs operating on the surface in terms of motion models, they are under different environmental communication conditions, and thus the AUV cooperative hunting problem has more constraints.”